# XEN^®^-63 Compared to XEN^®^-45 Gel Stents to Reduce Intraocular Pressure in Glaucoma

**DOI:** 10.3390/jcm12155043

**Published:** 2023-07-31

**Authors:** Charlotte Evers, Daniel Böhringer, Sara Kallee, Philip Keye, Heiko Philippin, Timothy Piotrowski, Thomas Reinhard, Jan Lübke

**Affiliations:** 1Eye Center, Medical Center—University of Freiburg, Faculty of Medicine, University of Freiburg, 79106 Freiburg, Germany; 2International Centre for Eye Health, Faculty of Infectious & Tropical Diseases, London School of Hygiene & Tropical Medicine, London WC1E 7HT, UK

**Keywords:** XEN^®^-63 gel stent, glaucoma, bleb revision

## Abstract

The XEN^®^ gel stent reduces intraocular pressure (IOP) in glaucoma. XEN^®^-45 is widely used; the newer XEN^®^-63 has a larger lumen targeting potentially lower IOP outcomes. We retrospectively compared the first 15 XEN^®^-63 cases to 15 matched XEN^®^-45 controls. With a preoperative IOP of 18.1 ± 3.9 mmHg (mean ± SD) and a final IOP of 9.1 ± 2.0 mmHg, XEN^®^-63 implantation resulted in an IOP reduction of 44.6 ± 16.5%. Similarly, with a preoperative IOP of 18.3 ± 4.5 mmHg and a final IOP of 10.3 ± 2.1 mmHg, XEN^®^-45 implantation resulted in an IOP reduction of 40.1 ± 17.2%. The median follow-up period was 204 days (range 78–338 days) for the XEN^®^-63 group and 386 days (range 99–1688 days) for the XEN^®^-45 group. In total, 5/15 eyes of each group underwent open conjunctival bleb revision within the period of observation. Three eyes of the XEN^®^-63 group had secondary glaucoma surgery. One eye in the XEN^®^-63 group and three eyes in the XEN^®^-45 group required a restart of antiglaucomatous medication. In conclusion, both stents effectively lower IOP and medication. XEN^®^-63 achieved a slightly lower IOP over a short follow-up. Complication and revision rates were similar.

## 1. Introduction

A XEN^®^ gel stent is a hollow cylindrical implant made of cross-linked collagen derived from porcine gelatin. Implanted ab interno through the sclera, the stent drains fluid from the anterior chamber to the subconjunctival space [1]. The XEN^®^-45 gel stent has been used successfully to treat primary open-angle glaucoma [2,3,4,5,6,7], as well as other forms of glaucoma [2,3,4,5,6,7], by reducing intraocular pressure (IOP) and the number of antiglaucomatous medications required. Compared to classic glaucoma filtration surgery, XEN^®^ implantation is less invasive and may have a better safety profile [8]. However, most studies show a high rate of needling [2,7,9,10,11] or bleb revision [4,12,13] after XEN^®^-45 implantation. In the long run, trabeculectomy can achieve lower IOP values [14]. XEN^®^-45 has a lumen of 45 µm and XEN^®^-63 a lumen of 63 µm. Both are 6 mm long and are implanted with a 27G injector. XEN^®^-45 has an outer diameter of 150 µm while XEN^®^-63 has an outer diameter of 170 µm. Real-world data for XEN^®^-45 implantation after 1–3 years show postoperative mean IOP levels of 14–16 mmHg [2,3,4,7,9,10,11,13,15]. Due to its lower outflow resistance compared to XEN^®^-45 (2–3 mmHg for XEN^®^-63 versus 6–8 mmHg for XEN^®^-45), XEN^®^-63 was designed to achieve a lower IOP level. So far, there are few studies on the current version of XEN^®^-63. Previously, studies reported on an earlier, non-marketed version with an inner diameter of 63 µm but an outer diameter of 240 µm, implanted with a 25G injector [16,17,18]. Fea et al. were the first to report on the newly marketed XEN^®^-63 [19,20]. Hussien et al. recently compared the novel 63 µm microstent to the conventional 45 µm microstent [21]. We herein present our data of XEN^®^-63 compared to XEN^®^-45 for different types of glaucoma in a tertiary center in Germany.

## 2. Materials and Methods

We conducted a retrospective analysis of the first 15 consecutive XEN^®^-63 implantations at our tertiary Eye Center at the University of Freiburg, Germany, compared to a matched group of XEN^®^-45 implantations. This study was performed in accordance with the principles of the Declaration of Helsinki. Approval was granted by the Ethics Committee of the University of Freiburg (No 21-1202_02).

Study patients: XEN^®^-63 gel stent (Allergan, an AbbVie company, Irvine, CA, USA) was implanted in 15 eyes of 13 patients with medically uncontrolled glaucoma. The control group consisted of 15 eyes of 15 patients who had received a XEN^®^-45 gel stent (Allergan, an AbbVie company, Irvine, CA, USA) implantation. Both groups were matched for age and type of glaucoma. Eyes with XEN^®^-45 implantation were selected from a quality control database using propensity matching based on age and type of glaucoma.

We obtained intraocular pressure using Goldmann applanation tonometry (GAT).

The surgical technique was as follows: Either XEN^®^-45 or XEN^®^-63 gel stent was implanted ab interno via a clear corneal incision without conjunctival dissection. Prior to the implantation, a very small amount of balanced salt solution was injected subconjunctivally in the target quadrant. At the end of the procedure, 4 mg of dexamethasone was administered intracamerally, and 0.1 mL of 0.2% mitomycin C was injected subconjunctivally. We did neither inject subconjunctival viscoelastic for bleb preparation nor leave any viscoelastic in the anterior chamber to prevent early hypotony. To reduce trauma to the conjunctiva, we did not perform conjunctival dissection in any of the primary procedures. Primary needling was performed if the subconjunctival portion of the XEN^®^ stent demonstrated restricted mobility. Postoperative treatment consisted of glucocorticoid eye drops (usually 1 mg/mL dexamethasone) 5 times daily, which were gradually reduced depending on the degree of postoperative conjunctival injection and intraocular inflammation (usually fortnightly reduction). There were no postoperative subconjunctival 5-fluorouracil injections given in any of the primary procedure cases. IOP-lowering drugs were stopped on the day of surgery.

Postoperative needling was performed in the case of a subconjunctivally encapsulated XEN^®^ stent, which at the discretion of the surgeon, could likely be released by needling alone. Criteria for bleb revision were signs of a dysfunctional bleb due to fibrotic tissue inhibiting the outflow through the subconjunctival part of the XEN^®^ stent or (impending) perforation of the XEN^®^ stent through the conjunctiva. A restart of topical antiglaucomatous treatment was resumed if the postoperative IOP exceeded the target level for that individual patient and the patient declined additional surgical interventions.

For open conjunctival bleb revision, we started with a conjunctival peritomy at the limbus followed by careful dissection of fibrotic tissue around the XEN^®^. After verification of good flow through the XEN^®^ stent, it was placed underneath Tenon’s fascia. Finally, the conjunctiva was closed using 7-0 vicryl sutures, and 0.1 mL of dexamethasone 4 mg/mL was injected subconjunctivally.

After postoperative needling and after bleb revision, patients received postoperative subconjunctival 5-fluorouracil injections (1 mL of 1% 5-fluorouracil) for three consecutive days following surgery and again glucocorticoid eye drops (usually 1 mg/mL dexamethasone) 5 times daily tapered fortnightly by one drop.

Analysis: The success of XEN^®^-63 versus XEN^®^-45 implantation was determined using descriptive statistics, including mean values, standard deviation, range, and median if differing substantially from the mean. Furthermore, we included Kaplan–Meier survival estimations. Criteria for failure in the Kaplan–Meier analysis were revision surgery with conjunctival dissection or secondary glaucoma intervention. To compare results between XEN^®^-63 and XEN^®^-45, an unpaired two-tailed *t*-test was performed.

We also analyzed IOP success rates according to the following criteria: Success was defined as a final IOP of ≤15 mmHg (≤12 mmHg) but ≥6 mmHg and an IOP reduction of ≥20% without (complete success) or with ocular hypotensive medications (qualified success) and without secondary glaucoma surgery. Failure was defined as an IOP level above the upper limit or below the lower limit or an IOP reduction < 20%. Complete failure was defined as a necessity for further glaucoma surgical intervention. Needling and bleb revision in this analysis were not recorded as evidence for failure.

## 3. Results

### 3.1. Patient Age and Type of Glaucoma

At the time of XEN^®^ implantation, patients in the XEN^®^-63 group were between 55 and 86 years old (mean age ± standard deviation: 75.3 ± 9.4 years), and patients in the XEN^®^-45 group were between 54 and 86 years old (mean age ± standard deviation: 74.8 ± 8.6 years).

The XEN^®^-63 group comprised five male patients (38.5%), two of whom received XEN^®^-63 implantation in both eyes and eight female patients (61.5%) who underwent XEN^®^-63 implantation in one eye. The XEN^®^-45 group comprised six male (40.0%) and nine female (60.0%) patients. Only one eye of each patient in this group was included in the study.

In both groups, 9/15 (60%) eyes had normal-tension glaucoma, 2/15 (13.3%) had primary open-angle glaucoma, 3/15 (20%) had pseudoexfoliation glaucoma, and 1/15 (6.6%) had uveitic glaucoma.

### 3.2. Preoperative IOP, Medications, and Previous Glaucoma Treatment

The mean preoperative IOP was 18.1 ± 3.9 mmHg (range 8–34 mmHg) in the XEN^®^-63 group and 18.3 ± 4.5 mmHg (range 10–35 mmHg) in the XEN^®^-45 group (no statistically significant difference, t = −0.12; df = 28; *p* = 0.9). The range of mean preoperative IOP per eye was 11–25 mmHg in the XEN^®^-63 group and 13–29.2 mmHg in the XEN^®^-45 group.

Preoperatively, all eyes in both groups were on topical hypotensive agents. Eyes in the XEN^®^-63 group were on an average of 3.3 substances, and eyes in the XEN^®^-45 group were on an average of 2.5 substances. In 2/15 cases (13.3%) in the XEN^®^-63 group and 1/15 cases (0.1%) in the XEN^®^-45 group, patients received additional oral acetazolamide in varying dosages.

In the XEN^®^-63 group, 6/15 eyes (40%) had previous glaucoma surgery, some with multiple interventions, including 4 Trabectome^®^ surgeries, 2 MINIject^®^ implantations, 2 cyclophotocoagulations, and 3x selective laser trabeculoplasties. The mean number of glaucoma procedures per eye in this group was 0.9. In case of previous cyclophotocoagulation, this was performed in the inferior quadrants and did not alter the condition of the superior conjunctiva where the bleb forms after XEN^®^ implantation. There were no cases of previous conjunctival incisional surgery.

In the XEN^®^-45 group, 4/15 eyes (27%) had previous glaucoma surgery. Two eyes had only selective laser trabeculoplasty. Another two eyes had selective laser trabeculoplasty and Trabectome^®^ surgery prior to XEN^®^-45 implantation.

In case of previous Trabectome^®^ surgery, gonioscopy showed the absence of the trabecular meshwork in the nasal iridocorneal angle, sometimes replaced by some scar tissue.

In both groups, 10 eyes were pseudophakic, and 5 eyes were phakic.

### 3.3. Postoperative Outcomes

The mean IOP within 4 days (mean 2.3 days) after XEN^®^-63 implantation was 7.7 ± 3.0 mmHg. IOP values within 4 days after XEN^®^-63 implantation ranged between 1–46 mmHg (range of mean IOP per eye 4.2–12.5 mmHg). XEN^®^-63 implantation resulted in a mean IOP reduction of 10.4 ± 3.8 mmHg (range 3.3–14.8 mmHg), or 56.5% (range 22.0–77.9%).

The mean IOP within 4 days (mean 2.6 days) after XEN^®^-45 implantation was 8.5 ± 3.1 mmHg. IOP values within 4 days after XEN^®^-45 implantation ranged between 1–39 mmHg (range of mean IOP per eye 3.3–13.1 mmHg). XEN^®^-45 implantation resulted in a mean IOP reduction of 9.7 ± 6.0 mmHg (range 0.9–22.6 mmHg), or 50.4% (range 6.4–86.6%).

Furthermore, immediate postoperative IOP-lowering medication could be discontinued in all eyes in both groups.

In 10/15 eyes (66.7%) in the XEN^®^-63 group, the postoperative IOP was <6 mmHg at least once, and in 4 eyes (26.7%), this lasted for over 1 week. In the XEN^®^-45 group, 8/15 eyes (53.3%) had a postoperative IOP < 6 mmHg, but in only 1 eye (0.07%) did this last for over 1 week.

There were no cases of choroidal effusion in the XEN^®^-63-group. In the XEN^®^-45 group, two cases showed transitional choroidal effusion immediately postoperatively associated with an IOP < 6 mmHg. In each group, among the patients with an IOP < 6 mmHg, there was one case of transient choroidal folds in the macular area.

Postoperative anterior chamber hemorrhage was observed in 8/15 eyes (53.3%) in the XEN^®^-63 group and 10/15 eyes (66.7%) in the XEN^®^-45 group. In most eyes, anterior chamber hemorrhage resolved within 2–3 weeks. In the XEN^®^-45 group, one of the eyes with anterior chamber hemorrhage had a transient IOP spike of 39 mmHg, and in another eye, resolution of the hyphema took five weeks. In both groups, three eyes with postoperative anterior chamber hemorrhage later underwent bleb revision. In the XEN^®^-63 group, one of these eyes also had secondary glaucoma surgery.

One patient in the XEN^®^-63 group had a suprachoroidal hemorrhage, which occurred on the first postoperative day. As a result, the IOP increased from 3 mmHg initially to a maximum of 46 mmHg, and antiglaucomatous treatment was restarted. Twelve days after surgery, choroidal drainage was performed in this patient. In addition, 40 days after surgery, the patient received secondary glaucoma intervention in the form of cyclophotocoagulation. Preoperative visual acuity was 20/32 or logMAR 0.22, and visual acuity at the last follow-up was 20/160 or logMAR 0.92 (133 days postoperatively). At the time of XEN^®^-63 implantation, this patient was 86 years old, and the eye undergoing XEN^®^ implantation had a diagnosis of pseudoexfoliation glaucoma for 18 years with a maximum IOP of 50 mmHg. The mean IOP on the day before surgery in this eye was 25.0 mmHg. Immediately postoperative, the IOP was 24 mmHg, but it dropped to 3 mmHg on the first postoperative day. The eye was pseudophakic and had previous myopia of −7 diopters. Before XEN^®^ implantation, the eye underwent selective laser trabeculoplasty, Trabectome^®^ surgery, as well as mild cyclophotocoagulation (inferior quadrants). The patient did not have other general diseases than a diagnosis of depression. In particular, the patient did not have arterial hypertension and was not on anticoagulants, antiplatelet drugs, or other cardiovascular drugs. The risk factors for suprachoroidal hemorrhage in this patient were the postoperative drop in IOP, resulting in a very low IOP, high myopia, long-term uncontrolled ocular hypertension glaucoma, previous intraocular surgeries, and the advanced age of the patient.

Suprachoroidal hemorrhage did not occur in any of the matched XEN^®^-45 eyes.

### 3.4. Follow-Up Period

The mean postoperative follow-up period was 199.1 ± 85.5 days (range 78–338 days; median 204 days) for the XEN^®^-63 group and 734.7 ± 619.8 days (range 99–1688 days; median 386 days) for the XEN^®^-45 group.

### 3.5. Post-XEN^®^ Interventions (Figure 1)

In the XEN^®^-63 group, 5/15 eyes (33.3%) underwent open conjunctival bleb revision an average of 45 days after surgery (range 28–74 days; median 39 days) due to impaired drainage from occlusion of the stent lumen with Tenon’s fascia or subconjunctival scarring. One of these patients had previous unsuccessful needling 6 days before bleb revision (50 days after XEN implantation). There was no primary needling in the XEN^®^-63 group.

**Figure 1 jcm-12-05043-f001:**
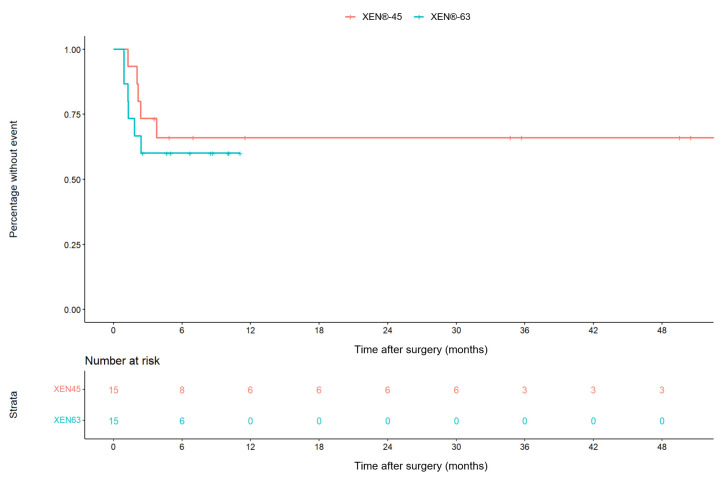
Kaplan–Meier curve for time to bleb revision or secondary glaucoma intervention over time. Steps indicate events, and ticks indicate eyes lost to follow-up. The follow-up period in this graphic is limited to 48 months. There was one additional open conjunctival bleb revision around 4.5 years after surgery in the XEN^®^-45 group. All other interventions occurred within 4 months of XEN^®^ implantation.

In the XEN^®^-45 group, the same percentage of eyes (33.3%) underwent open conjunctival bleb revision an average of 330 days after surgery (range 39–1624 days; median 70 days). Only one of these bleb revisions took place more than four months after XEN^®^-45 implantation (namely 1624 days or about 4.5 years) and was due to impending perforation of the conjunctiva by the XEN^®^ stent. In the XEN^®^-45 group, primary needling was performed in two eyes at the time of implantation. One of these eyes later underwent bleb revision 73 days postoperatively. Additionally, one eye in the XEN^®^-45 group had successful postoperative needling performed 63 days after implantation.

Apart from the aforementioned patient with suprachoroidal hemorrhage, who subsequently underwent cyclophotocoagulation, two other patients in the XEN^®^-63 group underwent PreserFlo^®^ surgery as secondary glaucoma intervention 32 days and 196 days after XEN^®^ implantation, respectively. Four days before PreserFlo^®^ implantation, one of these patients underwent XEN^®^ revision for an encapsulated bleb. After this bleb revision, IOP increased to 45 mmHg due to XEN^®^ obstruction by iris incarceration. The XEN^®^-63 implant was then replaced with a PreserFlo^®^. None of the XEN^®^-45 patients required secondary glaucoma intervention within the follow-up period.

### 3.6. Restart of Topical Treatment

No other patient in the XEN^®^-63 group besides the aforementioned one with suprachoroidal hemorrhage had antiglaucomatous treatment restarted within the follow-up period (three topical drugs plus oral acetazolamide).

In the XEN^®^-45 group, there was one patient who had to restart topical antiglaucomatous treatment (two agents) 302 days after XEN^®^ implantation or 263 days after bleb revision. Two other patients in the XEN^®^-45 group restarted topical antiglaucomatous treatment without previous bleb-revision 386 days (one agent) and 638 days (initially one agent, but later extended to four agents) post-surgery, respectively.

The mean number of topical antiglaucomatous agents at the end of follow-up was 0.2 ± 0.8 in the XEN^®^-63 group and 0.5 ± 1.1 in the XEN^®^-45 group. Compared to the baseline, the mean number of topical antiglaucomatous agents was reduced by 3.1 ± 0.9 agents (94.0%) in the XEN^®^-63 group and by 2.1 ± 1.8 agents (81.6%) in the XEN^®^-45 group.

### 3.7. IOP at Final Follow-Up

For IOP analysis at the final follow-up, we excluded all eyes with secondary glaucoma surgery. Eyes that underwent bleb revision but no secondary glaucoma surgery were included.

In the XEN^®^-63 group, the mean IOP at the final follow-up was 9.1 mmHg (range 6–12 mmHg). The mean IOP reduction was 7.9 mmHg (range 2.8–13.2 mmHg), or 44.6% (range 18.9–68.8) in the XEN^®^-63 group.

In the XEN^®^-45 group, the mean IOP at the final follow-up was 10.5 mmHg (range 7–15 mmHg). The mean IOP reduction was 7.8 mmHg (range 2.0–16.7 mmHg), or 39.8% (range 15.4–67.9%).

The following tables provide a summary of the main results for IOP values (Table 1), topical antiglaucomatous medication (Table 2), postoperative interventions (Table 3), and success rates (Table 4).

In the XEN^®^-63 group, 11/15 eyes (73.3%) showed complete success with a final IOP of not only ≤15 mmHg but also ≤12 mmHg and ≥6 mmHg as well as an IOP reduction of ≥20% without ocular hypotensive medications. One eye in the XEN^®^-63 group (6.7%) was censored as a failure despite having a final IOP of 12 mmHg because the IOP reduction from baseline was <20%. Three other eyes in the XEN^®^-63 group (20.0%) were censored as complete failure due to secondary glaucoma surgery.

In the XEN^®^-45 group, 10/15 eyes (66.7%) showed complete success with a final IOP of ≤15 mmHg and ≥6 mmHg, as well as an IOP reduction of ≥20% without ocular hypotensive medications. Only one of these eyes had a final IOP > 12 mmHg but an IOP reduction of >20%. Two eyes in the XEN^®^-45 group (13.3%) achieved qualified success with a final IOP ≤ 12 mmHg and ≥6 mmHg, as well as an IOP reduction of ≥20% with ocular hypotensive medications. Three other eyes in the XEN^®^-45 group (20.0%) were censored as a failure because the IOP reduction from baseline was <20%. The final IOP in these eyes was 15 mmHg, 13 mmHg, and 11 mmHg, respectively.

There was no statistically significant difference in baseline and final IOP or IOP reduction between the two groups (Table 5). This may be due to the small sample size.

## 4. Discussion

### 4.1. Effectivity

Our study shows effective IOP reduction after XEN^®^-63 implantation with mitomycin C. Compared to XEN^®^-45, the percentage IOP reduction at the end of follow-up was more pronounced (44.6% versus 39.8%), and the absolute IOP at the last visit was slightly lower (mean IOP of 9.1 mmHg versus 10.5 mmHg). These results are clinically relevant and favor XEN^®^-63, but due to the small sample size, they are not statistically significant. Furthermore, both XEN^®^-63 and XEN^®^-45 in our study showed high effectiveness in reducing medical treatment, with all but one patient off treatment at the end of follow-up in the XEN^®^-63 group and three patients having restarted topical treatment at the end of follow-up in the XEN^®^-45 group.

However, the follow-up period was short and differed between the two groups, with a median of 204 days (range 78–338 days) in the XEN^®^-63 group and a median of 386 days (range 99–1688 days) in the XEN^®^-45 group, which may influence the comparability of the two groups.

When interpreting our study results, it should also be considered that a high percentage of patients had normal-tension glaucoma. This distribution may limit how well the findings generalize to primary open-angle glaucoma populations. However, the fact that in a majority of cases, we observed substantial IOP reduction of ≥20% in a cohort dominated by normal-tension glaucoma suggests efficacy could potentially be even greater in cohorts comprising only primary open-angle glaucoma patients.

Fea et al. [19], in a retrospective study of XEN^®^-63 implantation with mitomycin C (n = 23) for primary open-angle glaucoma, found similar results after 3 months, with a slightly higher mean IOP (12.2 ± 3.4 mmHg) and slightly lower IOP reduction (40.8 ± 23.5%) at the end of follow-up (baseline IOP 27.0 ± 7.8 mmHg). They also found an effective reduction in the number of hypotensive medications. After 18 months [20], they reported a mean IOP of 14.1 ± 3.4 mmHg without hypotensive medication.

In their recently published retrospective cohort study, Hussien et al. [21] compared outcomes between XEN^®^-63 and XEN^®^-45 (*n* = 42 eyes of 41 patients per group) at 12 months post-surgery. They found a significantly higher complete success rate with XEN^®^-63 compared to XEN^®^-45 (59.5% vs. 28.6%, *p* = 0.009), although the qualified success rate was not significantly different between groups (66.7% vs. 45.2%, *p* = 0.08). The XEN^®^-63 group demonstrated a lower mean IOP (12.7 ± 4.8 vs. 15.5 ± 5.1 mmHg, *p* = 0.001) and required fewer medication classes (0.6 ± 1.1 vs. 1.7 ± 1.6 agents, *p* = 0.0005) compared to the XEN^®^-45 group. While the final IOP was lower in both of our study groups, our follow-up period was shorter compared to the study by Hussien et al. Additionally, unlike our study, they utilized both closed and open conjunctival approaches with ab interno XEN^®^ stent placement. Their cohort also included eyes with combined cataract surgery and previous subconjunctival surgery, which differed from our study population.

### 4.2. Safety

We observed more cases of hypotony < 6 mmHg lasting over one week after XEN^®^-63 implantation (4/15, 26.7%) compared to XEN^®^-45 implantation (1/15, 6.7%). Associated with hypotony, in the XEN^®^-45 group, 2/15 eyes (13.3%) showed transient choroidal effusion. In the XEN^®^-63 group, there was one case of suprachoroidal hemorrhage, a rare sight-threatening complication after XEN^®^ implantation, for which there are only a few case reports in the literature following or during XEN^®^-45 implantation [22,23,24].

Similarly, Hussien et al. [21] reported more distinct adverse events in the XEN^®^-63 group compared to XEN^®^-45 (34 vs. 19 events), though most were early and transient.

Fea et al. [19,20] reported a rate of 17.4% for both transient hypotony and choroidal detachment after XEN^®^-63 implantation.

### 4.3. Needling/Revision/Secondary Surgery

In both of our study groups, one-third of the eyes underwent open conjunctival bleb revision. In 9/10 eyes, this bleb revision was due to impaired drainage and occurred 2–4 months after XEN^®^ implantation. Therefore, impaired outflow in our study frequently occurred after implantation of both XEN^®^ types and quite early after XEN^®^ implantation (within 4 months). Only in one eye of the XEN^®^-45 group did it occur several years after implantation and was due to impending conjunctival perforation. Therefore, the short and differing follow-up periods between the two XEN^®^ groups may not be as relevant in this regard.

However, the shorter follow-up period for the XEN^®^-63 may artificially favor its outcome by providing insufficient time for failures or complications to manifest. The high rate of bleb revision required in both XEN^®^ groups may limit the success of this technique and should be considered when comparing XEN^®^ implantation to alternative glaucoma interventions.

A high revision rate has been reported for XEN^®^-45 in the literature [2,4,7,9,10,11,12,13]. To reduce postoperative needling and revision rates, open conjunctival or semi-open techniques have been proposed for XEN^®^ implantation [25,26,27]. For XEN^®^ implantation, we prefer an ab interno, closed conjunctiva approach, which it is designed for. In cases where an ab externo approach with open conjunctival dissection is favorable, we opt to use the PreserFlo^®^ micro shunt instead.

Secondary glaucoma intervention in this study was more frequent in the XEN^®^-63 group. However, this was due to rare complications that can also occur after XEN^®^-45 implantation. Given the small sample size, this difference cannot be considered evidence of a significant difference between the two XEN^®^ types.

In the study by Fea et al. [20], 17.4% underwent needling after a mean of 42.9 ± 11.2 days (one due to elevated IOP). Furthermore, 17.4% of their study required additional surgery: two trabeculectomies (8.7%), one XEN^®^ replacement with XEN^®^-45 (4.3%), one high-intensity focused ultrasound cyclodestruction (HIFU), and one more patient needed needling or additional surgery at the final follow-up.

There are several studies reporting promising results for needling or bleb revision after XEN^®^-45 implantation [28,29,30,31,32], some favoring bleb revision over needling [31,32], as we do in the case of bleb fibrosis. Studies on an older, non-marketed version of XEN^®^-63 after 1–5 years of follow-up reported an IOP reduction of 18–40% and a needling rate of 0–53% [16,17,18,33].

In the Hussien et al. study [21], the needling rate was 11.9% in each group done as an in-clinic intervention at the slit-lamp. The number of postoperative interventions (28 vs. 21, including needlings) and rate of reoperation (9 (21.4%) versus 6 (14.3%)) were higher in the XEN^®^-63 group compared to XEN^®^-45. Reoperations, considered failures in their study, included in-operating room revision surgeries.

### 4.4. Limitations

Our study is limited by its retrospective nature and relatively small sample size. The follow-up period was short and differed between the two groups. Our study may serve as an orientation for designing a prospective study comparing XEN^®^-63 versus XEN^®^-45.

## 5. Conclusions

XEN^®^-63 implantation with mitomycin C may lead to even lower IOP levels compared to XEN^®^-45 implantation over a short follow-up period. Larger studies with longer follow-ups are needed to confirm if this difference persists over time. The rates of complications and required revisions appear to be largely comparable. Further research is needed to evaluate the safety, efficacy, and role of XEN^®^-63 relative to XEN^®^-45 and other glaucoma procedures.

## Figures and Tables

**Table 1 jcm-12-05043-t001:** IOP values.

	XEN^®^-63 (15 Eyes of 13 Patients)		XEN^®^-45 (15 Eyes of 15 Patients)	
**At baseline:**	Mean	Range	Mean	Range
IOP	18.1 mmHg	8–34 mmHg	18.3 mmHg	10–35 mmHg
**Within 1–4 days**				
**after XEN^®^ implantation:**				
IOP	7.7 mmHg	1–46 mmHg	8.5 mmHg	1–39 mmHg
	range of mean IOP per eye: 4.2–12.5 mmHg		range of mean IOP per eye: 3.3–13.1 mmHg	
IOP reduction	10.4 mmHg	3.3–14.8 mmHg	9.7 mmHg	0.9–22.6 mmHg
	56.5%	22.0–77.9%	50.4%	6.43–86.6%
Follow-up period	199.1 ± 85.5 days (median 204 days)	78–338 days	734.7 ± 619.8 days (median 386 days)	99–1688 days
**At final follow-up:**				
(excluding patients with secondary glaucoma surgery)				
IOP	9.1 mmHg	6–12 mmHg	10.5 mmHg	7–15 mmHg
IOP reduction	7.9 mmHg	2.8–13.2 mmHg	7.8 mmHg	2–16.7 mmHg
	44.6%	18.9–68.8%	39.8%	15.4–67.9%

**Table 2 jcm-12-05043-t002:** Topical antiglaucomatous medication.

	XEN^®^-63 (15 Eyes of 13 Patients)	XEN^®^-45 (15 Eyes of 15 Patients)
**At baseline:**		
mean number of topical antiglaucomatous agents	3.3 agents	2.5 agents
eyes on topical treatment	15/15 eyes (100%)	15/15 eyes (100%)
**At final follow-up:**		
mean number of topical antiglaucomatous agents	0.2 agents	0.5 agents
eyes on topical treatment	1/15 eyes (6.7%)	3/15 eyes (20.0%)
reduction of topical treatment	−3.1 ± 0.9 (−93.9%)	−2.1 ± 1.8 (−81.6%)

**Table 3 jcm-12-05043-t003:** Postoperative interventions.

	XEN^®^-63 (15 Eyes of 13 Patients)		XEN^®^-45 (15 Eyes of 15 Patients)	
Postoperative needling	1/15 eyes	6.7%	1/15 eyes	6.7%
Bleb revision	5/15 eyes	33.3%	5/15 eyes	33.3%
Bleb revision within 6 months	5/15 eyes	33.3%	4/15 eyes	26.7%
Interval between XEN^®^ implantation and bleb revision	45 days (mean) 39 days (median)	range 28–74 days	330 days (mean) 70 days (median)	range 39–1624 days
Secondary glaucoma surgery	3/15 patients	20.0%	--	--
Interval between XEN^®^ implantation and secondary glaucoma surgery	89.3 days	range 32–196 days	--	--

**Table 4 jcm-12-05043-t004:** Success rates at final follow-up.

	XEN^®^-63 (15 Eyes of 13 Patients)		XEN^®^-45 (15 Eyes of 15 Patients)	
Complete success	11/15 eyes	73.3%	10/15 eyes	66.7%
(without ocular hypotensive medications, without secondary glaucoma surgery)				
Final IOP ≤ 15 mmHg ≥ 6 mmHg	12/15 eyes	80.0%	12/15 eyes	80.0%
Final IOP ≤ 12 mmHg ≥ 6 mmHg	12/15 eyes	80.0%	10/15 eyes	66.7%
IOP reduction ≥ 20%	11/15 eyes	73.3%	10/15 eyes	66.7%
**Qualified success**	--	--	**2/15 eyes**	**13.3%**
(with ocular hypotensive medications, without secondary glaucoma surgery)				
Final IOP ≤ 15 mmHg	--	--	3/15 eyes	20.0%
Final IOP ≤ 12 mmHg	--	--	2/15 eyes	13.3%
IOP reduction ≥ 20%	--	--	2/15 eyes	13.3%
**Failure**	**1/15 eyes**	**6.7%**	**3/15 eyes**	**20.0%**
(IOP reduction < 20%)				
**Complete Failure**	**3/15 eyes**	**20.0%**	--	--
(Secondary glaucoma surgery)				

**Table 5 jcm-12-05043-t005:** Results of unpaired *t*-test comparing IOP outcomes between XEN^®^-63 and XEN^®^-45.

	*t*	df	*p*
IOP at baseline	−0.12	28	0.90
IOP within a few days after XEN^®^ implantation	−0.74	28	0.47
IOP reduction within a few days after XEN^®^ implantation (total)	0.35	28	0.72
IOP reduction within a few days after XEN^®^ implantation (percentage)	0.87	28	0.39
IOP at final follow-up	−1.64	25	0.11
IOP reduction at final follow-up (total)	0.06	25	0.95
IOP reduction at final follow-up (percentage)	0.7	25	0.49

## Data Availability

The data presented in this study are available on request from the corresponding author. The data are not publicly available due to ethical restrictions.

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
