# Peer review of "XEN®-63 Compared to XEN®-45 Gel Stents to Reduce Intraocular Pressure in Glaucoma"

_jcm, 2023, doi:10.3390/jcm12155043_

Round 1
Reviewer 1 Report
Materials and Methods
Please define the success in terms of IOP, or optic nerve changes (an IOP cut off value, or percentage reduction from preoperative, for definition of success).
Results
Line 103 – 104: “In the XEN®-63 group, 6/15 eyes (40 %) had previous glaucoma surgery, some with 103 multiple interventions” 1. Please present the mean number of glaucoma surgeries per eye in this group. 2. Please mention the quadrant where cyclophotocoagulation (in the 2 eyes undergoing that procedure) was performed and the condition of the conjunctiva therein. 3. Please provide information on the gonioscopy findings in the study eyes having undergone previous angle procedures.
Line 112: “The mean IOP during the first few days” Please use a more precise description of time (e.g.: “in the first postoperative week”).
Line 113 (and others): “(range 4.2-12.5 mmHg).” Please mention the method of measurement of IOP (GAT measurements of IOP are whole numbers, not decimal points).
Lines 120 -123: “In 10/15 eyes (66.7%) in the XEN®-63 group, the postoperative IOP was < 6 mmHg ….., but in only one eye (0.07%) did this 122 last for over 1 week.” Please mention if any hypotony related events were encountered in these eyes (e.g. hypotony maculopathy, choroidal folds, optic nerve oedema, etc).
Line 124: “Postoperative anterior chamber hemorrhage” Please mention the clinical course of these cases over the follow up.
Line 126: “One patient in the XEN®-63 group had suprachoroidal hemorrhage” Please provide the full clinical details of this patient (age, co-morbidities, glaucoma diagnosis, phakic status, preoperative IOP, etc) given the gravity of the complication.
Line 135 – 136: “In each group, there was one case of transient choroidal 135 folds in the macular area.” Did these eyes match the IOP < 6 mmHg? Please mention.
Lines 139 – 141: Follow up period: for any glaucoma procedure, a reasonable follow up period is mandatory for accurate reporting. Follow ups of less than 3 months (90 days) are virtually non-informative. Hence, eyes with a few days of follow up (12 days, 18 days, etc) should be excluded from the study. Alternatively, data of these eyes may be presented in full alone to help the reader better understand the clinical course of such cases in their correct time context.
Reviewer 2 Report
General comments
· This study compares the surgical outcomes of XEN-63 with XEN-45 in terms of efficacy and safety profiles.
· The study is generally well written. However there are a few deficiencies in this study.
· One deficiency in this study sample size in each group is small therefore it is not clear if the study is sufficiently powered to examine differences between the safety profiles of the two devices. There is also a higher proportion of cases being normal tension glaucoma,with only 13% being POAG, thus limiting its generalised interpretation for POAG patients.
· Another deficiency is the relatively short followup for the groups, with average following up being only 57 days for XEN63 and 183 days for XEN 45.
· The difference in followup period also presents a problem when comparing the surgical outcomes of the two procedures. It is commonly know that failure of a glaucoma procedure can occur gradually over time, and if the followup period for XEN63 group is considerably shorter than XEN45 group then it is conceivable that the outcomes of the XEN63 group would be superior simply due to insufficient time for failure events and complication events to occur.
· There is also a high number of previous glaucoma surgeries in both groups, particularly the XEN63 group. It is not made clear if these cases had previous conjunctival incisional sugery. While Trabectome and Minijet is likely to not affect health of conjunctiva, cyclophotocoagulation can induce some degree of conjunctival scarring at previous sites of treatment.
· It is recommended that outcomes of the procedures should be reported as Success, Qualified success and Failure as per World Glaucoma Association reporting guidelines.
· Post operative needling rate should be reported.
· In the discussion section, authors should provide a balanced discussion and discuss the high rates of surgical failure in the form of bleb revision (33%) in both groups.
Specific comments
Materials and methods section
· authors should specify whether any bleb preparation steps were performed, such as primary needling or pre-insertion of viscoelastic. Authors should discuss why conjunctival dissection such as open or semi-open techniques were not used to avoid tenon obstruction to reduce post-operative needling and revision rates.
· authors should specify whether 5FU were given subconjunctivally post op in any of the primary procedure cases.
· authors should describe the procedure for open conjunctival bleb revision.
· Authors should specify criteria for needling, for restarting topical glaucoma drops or for bleb revision.
· Authors should specify whether measures to avoid early hypotony was implemented such as leaving small amount of viscoelastic in AC.
· For the single patient of XEN-63 group which developed suprachoroidal hemorrhage, author should include the case in the study summary, therefore the lines 112 – 113 should include upper IOP of 46mmHg.
Following period
· The mean following number of days for the two groups does not match that in the abstract. Is it 57 days for XEN 63 and 183 for XEN 45?
Post-XEN interventions section
· Authors should report on the overall needling rates of the two groups due to general high needling rate in XEN stents. Authors should also specify whether needling procedures were performed with post procedure 5FU or MMC.
IOP at final followup section
· Authors should indicate whether patients who had undergo revision were included in the final summary analysis.
Discussion session
· Lines 211-213 – While all but one patient is off treatment at the end of the trial period, the high proportion (33%) of cases from both groups required revision surgery should be considered the assessing the effectiveness of the procedures.
Round 2
Reviewer 2 Report
No further comments. All queries adequately addressed.
Author Response
Response to Reviewer 2
We are pleased to receive your feedback that all queries and comments on our manuscript, XEN®-63 compared to XEN®-45 gel stents to reduce intraocular pressure in glaucoma, have been satisfactorily addressed.
We appreciate the thoughtful review and constructive feedback provided, which has helped strengthen our work.